# STRUCTURAL LANDMARKING AND INTERACTION MODELLING: ON RESOLUTION DILEMMAS IN GRAPH CLASSIFICATION

## ABSTRACT

Graph neural networks are promising architecture for learning and inference with graph-structured data. However, generating informative graph level features has long been a challenge. Current practice of graph-pooling typically summarizes a graph by squeezing it into a single vector. However, from complex systems point of view, properties of a complex system are believed to arise largely from the interaction among its components. In this paper, we analyze the intrinsic difficulty in graph classification under the unified concept of "resolution dilemmas" and propose "*SLIM*", an inductive neural network model for Structural Landmarking and Interaction Modelling, to remedy the information loss in graph pooling. We show that, by projecting graphs onto end-to-end optimizable, and well-aligned substructure landmarks (representatives), the resolution dilemmas can be resolved effectively, so that explicit interacting relation between component parts of a graph can be leveraged directly in explaining its complexity and predicting its property. Empirical evaluations, in comparison with state-of-the-art, demonstrate promising results of our approach on a number of benchmark datasets for graph classification.

## 1 INTRODUCTION

Complex systems are ubiquitous in natural and scientific disciplines, and how the relation between component parts gives rise to global behaviour of a system is a central research topic in many areas such as system biology (Camacho et al., 2018), neural science (Kriegeskorte, 2015), and drug and material discoveries (Stokes et al., 2020; Schmidt et al., 2019). Recently, graph neural networks provide a promising architecture for representation learning on graphs – the structural abstraction of a complex system. State-of-the-art performances are observed in various graph mining tasks (Bronstein et al., 2017; Defferrard et al., 2016; Hamilton et al., 2017; Xu et al., 2019; Velickovic et al., 2017; Morris et al., 2019; Wu et al., 2020; Zhou et al., 2018; Zhang et al., 2020). However, due to the non-Euclidean nature, important challenges still exist in graph classification. For example, in order to generate a fixed-dimensional representation for a graph of arbitrary size, *graph pooling* is typically adopted to summarize the information from each each node. In the pooled form, the whole graph is squeezed into a "super-node", in which the identities of the constituent sub-graphs and their inter-connections are mixed together. Is this the best way to generate graph-level features? From a complex system's view, mixing all parts together might make it difficult for interpreting the prediction results, because properties of a complex system arise largely from the *interactions* among its components (Hartwell et al., 1999; Debarsy et al., 2017; Cilliers, 1998).

The choice of the "collapsing"-style graph pooling roots deeply in the lack of natural alignment among graphs that are not isomorphic. Therefore pooling sacrifices structural details for feature (dimension) compatibility. Recent years, substructure patterns[1] draw considerable attention in graph mining, such as motifs (Milo et al., 2002; Alon, 2007; Wernicke, 2006; Austin R. Benson, 2016) and graphlets (Shervashidze et al., 2009). It provides an intermediate scale for structure comparison or counting, and has been applied to node embedding (Lee et al., 2019; Ahmed et al., 2018), deep graph kernels (Yanardag & Vishwanathan, 2015) and graph convolution (Yang et al., 2018). However, due to combinatorial nature, only substructures of very small size (4 or 5 nodes) can be considered (Yanardag

---

[1]Informally, substructure in this paper means a connected subgraph and will be used interchargeably with it.

& Vishwanathan, 2015; Wernicke, 2006), greatly limiting the coverage of structural variations; also, handling substructures as discrete objects makes it difficult to compensate for their similarities, and so the risk of overfitting may rise in supervised learning scenarios (Yanardag & Vishwanathan, 2015).

We view these intrinsic difficulties as related to *resolution dilemmas* in graph-structured data processing. Resolution is the scale at which measurements can be made and/or information processing algorithms are conducted, and here we will discuss two types of resolution and related dilemmas: the spatial resolution (dilemma) and the structural resolution (dilemma).

*Spatial resolution* relates to the geometrical scale of the "component" that can be identified from the final representation of a graph (based on which the prediction is performed). In GNN, since graph pooling compresses the whole graph into a single vector, node and edge identities are mixed together and the spatial resolution drops to the lowest. We call this **vanishing spatial resolution (dilemma)**. *Structural resolution* is the fineness level in differentiating between two substructures. Currently practice of exact matching makes it computationally intractable to handle the exponentially many sub-graph instances, and the risk of overfitting may also rise as observed in deep graph kernels (Yanardag & Vishwanathan, 2015) and dictionary learning (Marsousi et al., 2014). We will call this over-delicate substructure profiling an **exploding structural resolution (dilemma)**. In fact, these two resolution dilemmas are not isolated. They have a causal relation and the origin is the way we perform identification and comparison of discrete substructures (more in Section 2.3).

**Our contribution**. Inspired by the well-studied science of complex systems, and in particular the importance of the interacting relation between component parts of a system, we propose a simple neural architecture called "Structural Landmarking and Interaction Modelling" - or SLIM. It allows graphs to be projected onto a set of end-to-end optimizable, well-aligned structural landmarks, so that identities of graph substructures and their interactions can be captured explicitly to explain the complexity and improve graph classification. We show that, by resolving the two resolution dilemmas, and subsequently respecting the structural organization of complex systems, SLIM can be empirically very promising and offers new possibilities in graph representation learning.

In the rest of the paper, we will first define the resolution dilemmas of graph classification in Section 2, together with the discussion of related works. We then cover in Section 3, 4 and 5 the design, analysis, and performance of SLIM, respectively. Finally, the last section concludes the paper.

## 2 RESOLUTION DILEMMAS IN GRAPH CLASSIFICATION

A complex system is often composed of many parts interacting with each other in a non-trivial way. Since graphs are structural abstraction of complex systems, accurate graph classification depends on how global properties of a system relate to its structure. It is believed that property of a complex system arises from interactions among its components (Debarsy et al., 2017; Cilliers, 1998). Consequently, accurate interaction modelling should benefit prediction. However, it is non-trivial due to resolution dilemmas, as described in the following subsections.

### 2.1 SPATIAL RESOLUTION DIMINISHES IN GRAPH POOLING

Graph neural networks (GNN) for graph classification typically involves two key blocks, graph convolution and graph pooling (Kipf & Welling, 2017; Hamilton et al., 2017; Xu et al., 2019), at significantly different spatial resolutions. The goal of convolution is to pass information among neighboring nodes in the general form of

$$h_v = \text{AGGREGATE}\left(\{h_u, u \in \mathcal{N}_v\}\right),$$

where $\mathcal{N}_v$ is the neighbors of $v$ (Hamilton et al., 2017; Xu et al., 2019). Here, the spatial resolution is controlled by the number of convolution layers: more layers capture lager substructures/sub-trees and can lead to improved discriminative power (Xu et al., 2019). In other words, the spatial resolution in the convolution stage can be controlled easily, and multiple resolutions may be even combined together via CONCATENATE function (Hamilton et al., 2017; Xu et al., 2019) for improved modelling.

The goal of graph pooling is to generate compact, graph- or subgraph-level representations that are compatible across graphs. Due to the lack of natural alignment between non-isomorphic graphs, graph pooling typically "squeezes" a graph $\mathcal{G}$ into a single vector (or "super-node") in the form of

$$h_\mathcal{G} = \text{READOUT}\left(\{f(h_v), \forall v \in \mathcal{V}\}\right),$$

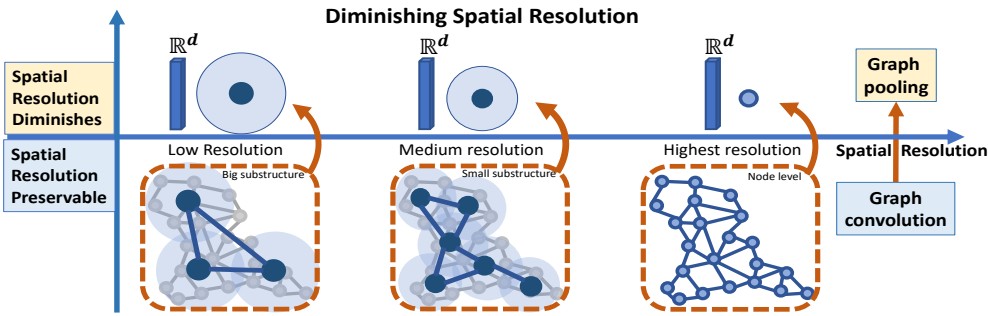

Figure 1: Spatial resolution tunable in convolution phase (bottom, with 3 resolution level examples illustrated), but vanishes in the pooling phase (top, which mixes all the node vectors into one vector).

where $\mathcal{V}$ is the vertex set of $\mathcal{G}$. The readout functions include: max-pooling (Cangea et al., 2018), sum-pooling (Xu et al., 2019), some other pooling functions (Hamilton et al., 2017), or deep sets (Zaheer et al., 2017); attention has been used to evaluate node importance in attention pooling (Lee et al., 2019) and gPool (Gao & Ji, 2019); hierarchical pooling has also been investigated (Ying et al., 2018). The spatial resolution drops significantly in graph pooling, as shown in Figure 1. Since all the nodes (and their representation) are mixed into one vector, subsequent classifier can no longer identify any individual substructure regardless of the spatial resolution used in the convolution stage. We call this "diminishing spatial resolution". A diminishing spatial resolution will mix the identity of sub-structures (e.g., functional modules of a molecule), making it non-trivial to trace the behaviour of the classifier back to meaningful parts of the graph for interpretation.

## 2.2 STRUCTURAL RESOLUTION EXPLODES IN SUBSTRUCTURE IDENTIFICATION

Substructures are the basic unit to accommodate interacting relations. A global criterion to identify and align substructures is the key to preserving substructure identities and comparing the inherent interactions across graphs. Again, the granularity in determining whether two substructures are "similar" or "different" is subject to a wide spectrum of choices, which we call "structural resolution".

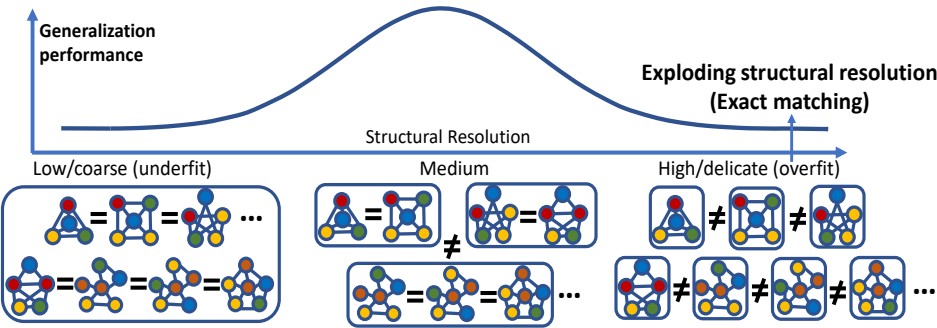

Figure 2: Over-delicate structural resolution may hamper the generalization performance.

We illustrate the concept in Figure 2. The right end denotes the finest resolution in differentiating substructures: exact matching, as we manipulate motif/graphlet (Milo et al., 2002; Alon, 2007; Wernicke, 2006; Yang et al., 2018; Shervashidze et al., 2009). The exponential configuration of sub-graphs will finally lead to an "exploding" structural resolution, because maintaining a large number of unique substructures is computationally infeasible, and easily overfits (Yanardag & Vishwanathan, 2015). The left end of the spectrum treats all substructures the same and underfits the data. We are interested in a medium structural resolution, where similar substructures are mapped to the same identity, which we believe can benefit the generalization performance (empirical evidence in Fig. 5).

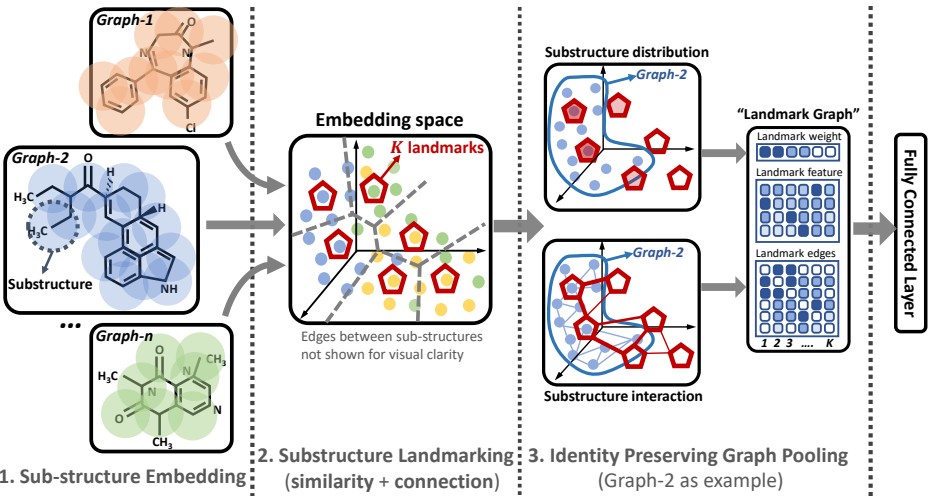

Figure 3: Three steps of the SLIM network. Sub-structure embedding: extract $k$-hop local sub-graphs around each node and embed them in a new space. Substructure landmarking: compute sub-structure representatives through unsupervised clustering of sub-structures across graphs. Identity-preserving graph pooling: project each graph on the common set of sub-structure landmarks for final prediction.

## 2.3 RELATION BETWEEN SPATIAL AND STRUCTURAL RESOLUTION DILEMMAS

The two resolution dilemmas have a causal relation, and the logic chain is as follows.

1. Due to difficulty of characterizing discrete subgraphs, **exact matching** is typically adopted.
2. As a result, an **exploding structural resolution** (dilemma) is caused.
3. Such an over-delicate granularity makes it infeasible to compare substructure across graphs.
4. As a result, a **collapsing-style graph pooling** has to be adopted that summarizes the whole graph into a single vector, serving as compatible graph-level features.
5. As a result, a **vanishing spatial resolution** (dilemma) caused finally.

Namely, the exploding structural resolution makes (collapsing-style) graph pooling an inevitable choice, which in turn leads to diminishing spatial resolution. Since the root cause of exploding structural resolution is how we typically manipulate discrete sub-structures, i.e., exact matching, we will replace it with structural landmarking in SLIM, so that both dilemmas are coordinately solved.

## 3 STRUCTURAL LANDMARKING AND INTERACTION MODELLING (SLIM)

The key idea of SLIM is to compute landmarks (or representatives) from the distribution of sub-structures (embedded in a continuous space) across different graphs. By doing this, identification and comparison of sub-structures become much easier, and so an identity-preserving graph pooling becomes applicable that explicitly models interaction between component parts of a graph.

**Problem Setting**. Give a set of labeled graphs $\{\mathcal{G}_i, y_i\}$'s for $i = 1, 2, ..., n$, with each graph defined on the node/edge set $\mathcal{G}_i = (\mathbf{V}_i, \mathbf{E}_i)$ with adjacency matrix $\mathbf{A}_i \in \mathbb{R}^{n_i \times n_i}$ where $n_i = |\mathbf{V}_i|$, and $y_i \in \{\pm 1\}$. Assume that nodes are drawn from $c$ categories, and the node attribute matrix for $\mathcal{G}_i$ is $\mathbf{X}_i \in \mathbb{R}^{n_i \times c}$. Our goal is to train an inductive model to predict the labels of the testing graphs.

The SLIM network has three main steps: (1) substurcture embedding, (2) substructure landmarking, and (3) identity-preserving graph pooling, as shown in Figure 3. Detailed discussion follows.

### 3.1 SUBSTRUCTURE EMBEDDING

The goal of substructure embedding is to extract substructure instances and embed them in a metric space. One can employ multiple layers of graph convolution (Hamilton et al., 2017; Xu et al., 2019)

to model substructures (in fact, rooted sub-trees growing from each node). In Figure 3, sub-graphs in each shaded circle represents a substructure instance associated with one atom.

More specifically, we extract one sub-graph instance from each node using an $h$-hop breadth-first search. Let $\mathbf{A}_i^{(k)}$ be the $k^{th}$-order adjacency matrix, i.e., the $pq^{th}$ entry equals 1 only if node $p$ and $q$ are within $k$-hops away. Since each sub-graph is associated with one node, the sub-graphs extracted from $\mathcal{G}_i$ can be represented as $\mathbf{Z}_i = \mathbf{A}_i^{(k)} \mathbf{X}_i$, whose $j^{th}$ row is a $c$-dimensional vector summarizing counts of the $c$ node-types in the sub-graph around the $j^{th}$ node. Again, different variations of graph convolution (Kipf & Welling, 2017; Hamilton et al., 2017) can be adopted (see Appendix A).

Next we consider embedding the substructure instances (i.e., rows of $\mathbf{Z}_i$'s) from each graph jointly into a latent space so that statistical manipulations become feasible. The embedding should preserve important proximity relations to facilitate subsequent landmarking: if two substructures are similar, or they frequently inter-connect with each other, their embeddings should be close. In other words, the embedding should be smooth with regard to both structural similarities and geometrical interactions.

A parametric transform on $\mathbf{Z}_i$'s with controlled complexity can guarantee the smoothness of embedding w.r.t. structural similarity, e.g., an autoencoder (with one hidden-layer as example)

$$f(\mathbf{Z}_i) = \sigma\left(\mathbf{Z}_i \cdot \mathbf{T} + \mathbf{b}\right). \tag{1}$$

Here $\mathbf{T}$ and $b$ are the transform matrix and bias term of the autoencoder. Let $\mathbf{H}_i = f(\mathbf{Z}_i) \in \mathbb{R}^{n_i \times d}$ be the embedding of the $n_i$ sub-graph instances extracted from $\mathcal{G}_i$. To maintain the smoothness of $\mathbf{H}_i$'s w.r.t. geometric interaction, we maximize the log-likelihood of the co-occurrence of substructure instances in each graph, similar to word2vec (Mikolov et al., 2013)

$$\max \sum_{i=1}^{n} \sum_{j=1}^{n_i} \sum_{l \in \mathcal{N}_j^i} \log \left( \frac{\exp\langle \mathbf{H}_i(j,:), \mathbf{H}_i(l,:)\rangle}{\sum_{l'} \exp\langle \mathbf{H}_i(j,:), \mathbf{H}_i(l',:)\rangle} \right) \tag{2}$$

Here $\mathbf{H}_i(j,:)$ is the $j^{th}$ row of $\mathbf{H}_i$, $\langle,\rangle$ is inner product, and $\mathcal{N}_j^i$ are the neighbors of node $j$ in graph $\mathcal{G}_i$. This loss function tends to embed strongly inter-connecting substructures close to each other.

## 3.2 SUBSTRUCTURE LANDMARKING

The goal of structural landmarking is to identify a set of informative structural landmarks in the continuous embedding space such that these landmarks have: (1) high statistical coverage, namely, they should faithfully recover distribution of the substructures from the input graphs, so that we can generalize them to new substructure examples from the distribution; and (2) high discriminative power, namely the landmarks should be able to reflect discriminative interaction patterns for classification.

Let $\mathbf{U} = \{\boldsymbol{\mu}_1, \boldsymbol{\mu}_2, ..., \boldsymbol{\mu}_K\}$ be the set of structural landmarks. In order for them to be representative of the substructures from the input graphs, it is desirable that each sub-graph instance can be faithfully approximated by the closest landmark in $\mathbf{U}$. Thus, we minimize the following distortion loss

$$\sum_{i=1}^{n} \sum_{j=1}^{n_i} \min_{k=1,2,...,K} \|\mathbf{H}_i(j,:) - \boldsymbol{\mu}_k\|^2. \tag{3}$$

Here $\mathbf{H}_i(j,:)$ denotes the $j^{th}$ row (substructure) from graph $\mathcal{G}_i$. In practice, we implement soft assignment by using a cluster indicator matrix $\mathbf{W}_i \in \mathbb{R}^{n_i \times k}$ for each graph $\mathcal{G}_i$, whose $jk^{th}$ entry is the probability that the $j^{th}$ substructure of $\mathcal{G}_i$ belongs to the $k^{th}$ landmark $\boldsymbol{\mu}_k$. Inspired by deep embedding clustering (Xie et al., 2016), $\mathbf{W}_i$ is parameterized by a Student's t-distribution

$$\mathbf{W}_i(j,k) = \frac{(1 + \|\mathbf{H}_i(j,:) - \boldsymbol{\mu}_k\|^2/\alpha)^{-\frac{\alpha+1}{2}}}{\sum_{k'}(1 + \|\mathbf{H}_i(j,:) - \boldsymbol{\mu}'_k\|^2/\alpha)^{-\frac{\alpha+1}{2}}},$$

and the loss function can be greatly simplified by minimizing the KL-divergence

$$\min_{\mathbf{U},\mathbf{H}'_i\text{s}} \sum_i \mathrm{KL}\left(\mathbf{W}_i, \widetilde{\mathbf{W}}_i\right), \quad \text{s.t.} \quad \widetilde{\mathbf{W}}_i(j,k) = \frac{\mathbf{W}_i^2(j,k)/\sum_l \mathbf{W}_i(l,k)}{\sum_{k'}[\mathbf{W}_i^2(j,k')/\sum_l \mathbf{W}_i(l,k')]}. \tag{4}$$

Here, $\widetilde{\mathbf{W}}_i$ is a self-sharpening version of $\mathbf{W}_i$, and minimizing the KL-distance forces each substructure instance to be assigned to only a small number of landmarks similar to sparse dictionary learning. Besides the unsupervised regularization in (3) or (4), learning of the structural landmarks is also affected by the classification loss, guaranteeing the discriminative power of the landmarks.

### 3.3 IDENTITY-AND-INTERACTION-PRESERVING GRAPH POOLING

The goal of identity-and-interaction-preserving graph pooling is to project structural details of each graph onto the common space of landmarks, so that a compatible, graph-level feature can be obtained that simultaneously preserves the identity of the parts (substructures) and models their interactions.

The structural landmarking mechanism allows computing rich, compatible graph-level features.

1. Substructure distribution in each graph. The density of the $K$ substructure landmarks in graph $\mathcal{G}_i$ can be computed as

$$\mathbf{p}_i = \mathbf{W}'_i \cdot \mathbf{1}_{n_i \times 1}. \tag{5}$$

2. First-order moment of substructures in each graph. The mean of the substructures belonging to each of the $K$ landmarks in graph $\mathcal{G}_i$ is

$$\mathbf{M}_i = \mathbf{X}'_i \cdot \mathbf{W}_i \cdot \mathbf{P}_i^{-1} \tag{6}$$

with $\mathbf{P}_i = \text{diag}(\mathbf{p}_i)$: $k$th column of $\mathbf{M}_i$ is the mean of sub-graphs with the $k$th landmark.

3. Sub-structure interaction in each graph. We can model how the $K$ landmarks interact with each other in graph $\mathcal{G}_i$. To do this, we can project the adjacency matrices $\mathbf{A}_i$'s onto the landmark sets and obtain an $\mathbb{R}^{K \times K}$ interaction matrix

$$\mathbf{C}_i = \mathbf{W}_i \cdot \mathbf{A}_i \cdot \mathbf{W}'_i, \tag{7}$$

which encodes the interacting relations (geometric connections) among the $K$ structural landmarks. We can further normalize this interaction as $\tilde{\mathbf{C}}_i = \mathbf{P}_i^{-1}\mathbf{C}_i\mathbf{P}_i^{-1}$.

These features can be used together for final classification. For example, they can be concatenated to feed into fully-connected layer. One can also combine them together and transform each graph $\mathcal{G}_i$ into a constant-sized "landmark" graph with node feature $\mathbf{M}_i$, node weight $\mathbf{p}_i$, and edge weights $\mathbf{C}_i$. Then the standard graph convolution can be applied to the landmark graphs to generate graph-level features (without pains of graph alignment anymore). In experiments, we simply use normalized sub-structure interaction matrix $\tilde{\mathbf{C}}_i$ (re-shaped to vector) as graph-level feature (more details in Appendix H).

We illustrate the architecture of SLIM in Figure 4. Here, the final graph features are fed into one fully-connected layer with dimension 64 for final prediction. Globally, the objective function includes the supervised part, namely the cross-entropy loss for classification, and the unsupervised part, namely the node2vec loss (2) reflecting geometric connections between substructures within each graph, and the clustering loss (4) reflecting the sub-structure similarity across different graphs. Weights for unsupervised loss terms in (2) and (4) are 1 and 10, respectively. More details are in Section 5.

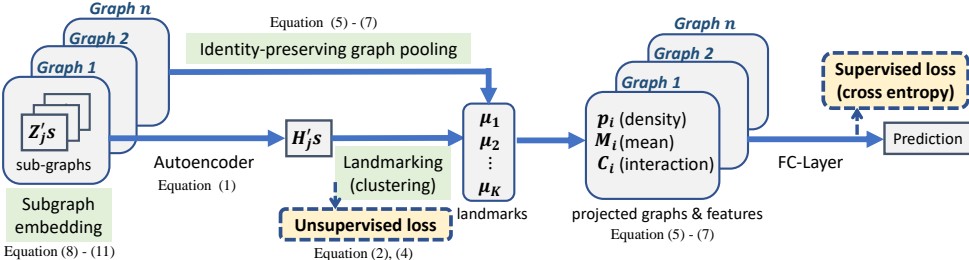

Figure 4: End-to-end training architecture of the SLIM network.

## 4 DISCUSSIONS

### 4.1 STRUCTURAL RESOLUTION FROM CODING PERSPECTIVE

The structural resolution is controlled by the landmark size $K$, which can be deemed as the number of basis in dictionary learning. Note that graphs can be considered as constructed from (inter-connected)

substructure instances, each instance $\mathbf{z}$ represented by the landmarks as $\mathbf{z} = \sum_k \alpha_k \boldsymbol{\mu}_k$. In other words, the structural landmarks can be deemed as code vectors (or basis) in a dictionary.

In dictionary learning, it is known that in general neither too small nor too large dictionary is desirable. A too small number of code-vectors fail to recover basic data structures, whereas too many basis may result in overfitting (Marsousi et al., 2014). In particular, when the redundancy of the basis vectors, as measured by the "mutual coherence", exceeds a certain range, the probability of a faithful signal recovery diminishes due to instability of sparse coding (Donoho & Huo., 2001; Mehta & Gray, 2013).

We have very similar observations on the choice of structural resolution $K$. Note that in case of exact sub-structure matching, which corresponds to a maximal $K$ due to the combinitorial nature of substructures, the redundancy in structural landmarks increases significantly, which violates the recovery condition and leads to inferior performance. This is why we prefer a reasonable control on the structural resolution, in contrast to exact sub-structure matching as in graphlets or motif discovery. In the Appendix B (Theorem 1), we provide a more detailed theoretic analysis on how the mutual coherence of the structural landmarks depends on the choice of structural resolution $K$.

## 4.2 Comparison with Related Methods

**Graph Isomorphism**. GNNs have great potential in graph isomorphism test by generating injective node embedding, thanks to the theoretic foundation in (Xu et al., 2019; Morris et al., 2019). SLIM provides new insight here: (1) it finds a tradeoff in handling similarity and distinctness; while graph isomorphism network (Xu et al., 2019) tries to discern even the slightest difference of sub-structures, SLIM network groups similar sub-structures (using tunable granularity), so that graphs can be projected onto structural landmarks; (2) it explores new ways of generating graph-level features: instead of aggregating all parts together, it taps into the vision of complex systems so that interaction relation is leveraged to explain the complexity and improve the learning (more in Appendix I).

**Graphlets, Graph Kernels, and Embedding Methods**. Graphlets and Graph kernels both exploit substructures to characterize graphs and their similarity. The SLIM network has some key differences. First, we consider sub-structure landmarks that are **end-to-end optimizable** for generating discriminative, graph-level interacting patterns, while graph kernels or graphlets typically enumerate sub-structures offline. Second, SLIM models interaction between sub-structures, which is very different from graph kernels. Recently, Deep Divergence Graph Kernel (Al-Rfou et al., 2019) is proposed for unsupervised graph similarity learning. The authors designed isomorphism attention mechanism to compare nodes from source and target graphs, while SLIM uses global clustering across a batch of (training) graphs. Finally, embedding-type methods mostly focus on node- or edge-level representations (Mikolov et al., 2013; Ahmed et al., 2018)(see more in Appendix J).

**Other Aggregation or Pooling Methods**. Hierarchical pooling (Ying et al., 2018; Gao & Ji, 2019; Lee et al., 2019; Khasahmadi et al., 2020) can exploit non-flat graph organization, but the final output is still in the form of a single, aggregated node vector. Sortpooling re-arranges graph nodes in a linear chain and perform 1d-convolution (Zhang et al., 2018); SEED uses distribution of multiple random walks to capture graph structures (Wang et al., 2019); Deep graph kernel evaluates graph similarity by subgraph counts (Yanardag & Vishwanathan, 2015). Explicit modelling of the interactionrelation in graphs is still not considered in these approaches.

## 5 Experiments

**Benchmark data**. We used the following benchmark data sets. (1) MUTAG: chemical compound with 188 instances and two classes; there are 7 node/atom types. (2) PROTEINS: protein molecules with 1,113 instances and three classes/node-types. (3) NCI1: chemical compounds for cancer cell lines with 4,110 instances and two classes. (4) PTC: chemical compounds for toxicology prediction with 417 instances and 8 classes. (5) D&D: enzyme classification with 1,178 instances and two classes. (6) IMDB-B: movie collaboration data set with 1,000 instances and two classes. (7) IMDB-M: movie collaboration data set with 1,500 instances and three classes.(8) COLLAB: scientific collaboration network from 3 physics fields (classes). We list the detailed statistics of the graph data sets in Table 1.

Table 1: Statistics of the benchmark data-sets (bioinformatics & social-net).

| Data | Num graphs | Classes | Node labels | Avg. nodes | Avg. edges | Class Ratio |
|------|-----------|---------|-------------|-----------|-----------|-------------|
| MUTAG | 188 | 2 | 7 | 17.93 | 19.79 | 1 : 1.9 |
| PTC | 344 | 2 | 19 | 14.29 | 14.69 | 1 : 1.2 |
| NCI1 | 4110 | 2 | 37 | 29.87 | 32.30 | 1 : 1.0 |
| Protein | 1,113 | 2 | 3 | 39.06 | 72.82 | 1 : 1.4 |
| DD | 1,178 | 2 | 82 | 284.32 | 715.66 | 1 : 1.4 |
| IMDB-B | 1000 | 2 | 1 | 19.77 | 96.53 | 1 : 1.0 |
| IMDB-M | 1500 | 3 | 1 | 13.0 | 65.94 | 1 : 1.0 : 1.0 |
| COLLAB | 5000 | 3 | 1 | 74.49 | 2457.78 | 1 : 2.1 : 3.3 |

**Competing methods**. We considered (1) Graph neural tangent kernel (GNTK) (Du et al., 2019); (2) Graph Isomorphism Network (GIN) (Xu et al., 2019); (3) End-to-end graph classification (DCGNN) (Zhang et al., 2018); (4) Hierarchical and differential pooling (DiffPool) (Ying et al., 2018); (5) Self-attention Pooling (SAG) (Lee et al., 2019); (6) Convolutional network for graphs (PATCHY-SAN) (Niepert et al., 2016); (7) Graphlet kernel (GK) (Shervashidze et al., 2009); (8) Weisfeiler-Lehman Graph Kernels (WLGK) (Shervashidze et al., 2011); and (9) Propagation kernel (PK) (Neumann et al., 2016). For method (4),(6),(7),(8),(9) we directly cited their reported results due to unavailability of their codes; for other competing methods we run their codes and report the results.

**Experimental setting**. Following setting in (Xu et al., 2019) and (Niepert et al., 2016), we evenly split the data into 10 folds and report the average and standard deviation of the accuracies of the 10 rounds. Our spatial resolution is controlled by a BFS with 3 hops; the structural resolution is set to $K = 100$. Hyper-parameter selection is based on small portion of the training splits as validation set, including: (1) the number of hidden units in the autoencoder Eq.(1) is chosen from $\{d, d/2, 2d, 8, 16\}$ where $d$ is the input dimension of the encoder; (2) the optimizer is chosen from SGD and Adagrad, with learning rate $\{1e-2, 5e-2, 1e-3, 5e-3, 1e-4\}$; (3) local graph representation, including node distribution, layer-wise distribution, and weighted layer-wise summation (more in Appendix A).

**Structural Resolution**. In Figure 5, we examine the performance of SLIM with different structural resolution ($K$). As can be seen, the accuracy curve is bell-shaped. When $K$ is either too small (underfitting) or too large (coherent landmarks that overfit), the accuracy is inferior, and the best performance is typically obtained around a median $K$ value. This justifies our conjecture, as well as the usefulness of structural landmarking in improving graph classification.

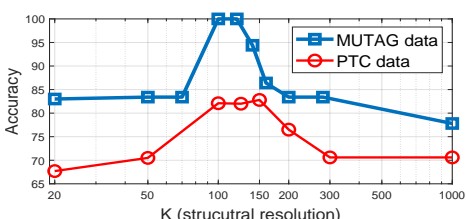

Figure 5: Accuracy vs structural resolution $K$.

**Classification Performance**. We compare the performance of different methods in Table 2. We use three social network data sets (IMDB-B, IMDB-M, COLLAB) and five bioinformatics data sets (MUTAG, PTC, NCI1 PROTEINS, DD). As can be seen, in Table 2, overall, neural network-based approaches are more competitive than graph kernels, except that graph kernels have lower fluctuations, and the WL-graph kernel perform the best on the NCI1 dataset. For social network data, the SLIM network gains a competitive score on IMDB-B and IMDB-M, but is worse on the COLLAB. We speculate that social network data do not have node features and so the advantage of SLIM might be less significant. From Table 3, SLIM yields the highest average ranking among all the 8 benchmark datasets.

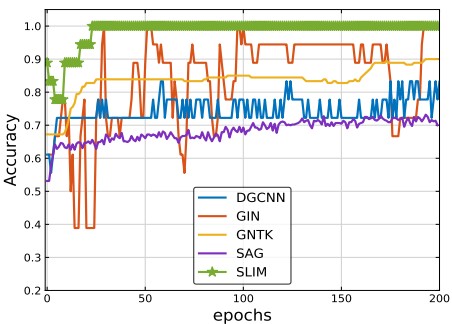

Figure 6: Algorithm stability on the `Mutag` dataset. **More examples in Appendix G**.

**Algorithm Stability**. In Figure 6 (more results in Appendix G.1, Figure 9), we plot the evolution of the testing accuracy versus the training epochs, so as to have a more comprehensive evaluation on algorithm stability. As can be seen, our approach has an accuracy curve that converges relatively faster

Table 2: Averaged algorithm performance on 8 benchmark data-sets (bioinformatics & social-net).

| ALG. | MUTAG | PTC | NCI1 | Protein | D&D | IMDB-B | IMDB-M | COLLAB |
|---|---|---|---|---|---|---|---|---|
| GK | 81.38±1.74 | 55.65±0.46 | 62.49±0.27 | 71.39±0.31 | 74.38±0.69 | 65.87±0.98 | 43.89±0.38 | 72.84±0.28 |
| PK | 76.00±2.69 | 59.50±2.44 | 82.54±0.47 | 73.68±0.68 | 78.25±0.51 | — | — | — |
| WLGK | 84.11±1.91 | 57.97±2.49 | **84.46±0.45** | 74.68±0.49 | 78.34±0.62 | 73.40±4.63 | 49.33±4.75 | 79.02±1.77 |
| PC-SAN | 92.63±4.21 | 60.00±4.82 | 78.59±1.89 | 75.89±2.76 | 77.12±2.41 | 71.00±2.29 | 45.23±2.84 | 72.60±2.15 |
| DGCNN | 85.83±1.66 | 58.59±2.47 | 74.46±0.47 | 75.54±0.94 | 79.37±1.03 | 70.03±0.86 | 47.83±0.85 | 73.76±0.49 |
| DiffPool | 90.52±3.98 | — | 76.53±2.23 | 75.82±3.56 | 78.95±2.40 | 78.08±4.24 | 53.13±4.70 | 79.70±1.84 |
| GNTK | 90.12±8.58 | 67.92±6.98 | 75.20±1.53 | 75.61±4.24 | **79.42±2.18** | 76.90±3.60 | 52.80±4.60 | **83.60±1.22** |
| SAG | 73.53±9.68 | 69.67±3.12 | 74.18±1.29 | 71.86±0.97 | 76.91±2.12 | 78.10±4.20 | 53.80±4.08 | 79.88±1.02 |
| GIN | 90.03±8.82 | 64.60±7.00 | 79.84±4.57 | 71.28±2.65 | 77.58±2.94 | 75.10±5.10 | 52.30±2.80 | 80.21±1.92 |
| SLIM | **93.28±3.36** | **74.41±6.92** | 80.53±2.01 | **77.47±4.34** | 79.61±2.66 | **80.20±2.10** | **54.00±4.02** | 78.22±2.02 |

Table 3: Averaged ranking of different algorithms on the 8 benchmark data-sets.

| ALG. | GK | PK | WLGK | PC-SAN | DGCNN | DiffPool | GNTK | SAG | GIN | SLIM |
|---|---|---|---|---|---|---|---|---|---|---|
| Rank | 8.7 | 6.4 | 5.6 | 5.8 | 5.8 | 4.1 | 3.7 | 4.7 | 4.6 | **1.9** |

and remains more stable with respect to the epochs. This signifies a small variance during the training process and makes it practically easy to determine when to stop training. Other GNN algorithms can also attain a high accuracy on some of the benchmark datasets, but the prediction performance fluctuates significantly across the training epochs (even by using very large mini-batch sizes). In such cases, determining when to stop can be challenging. We speculate that stability of the SLIM network arises from explicit modelling of the sub-structure distributions. In Figure 6, it is also worthwhile to note that on the MUTAG data the proposed method produces a classification with 100% accuracy on more than half of the runs across different folds, and converges to the perfect classification steadily. It demonstrates the power of the SLIM network in capturing important graph-level features.

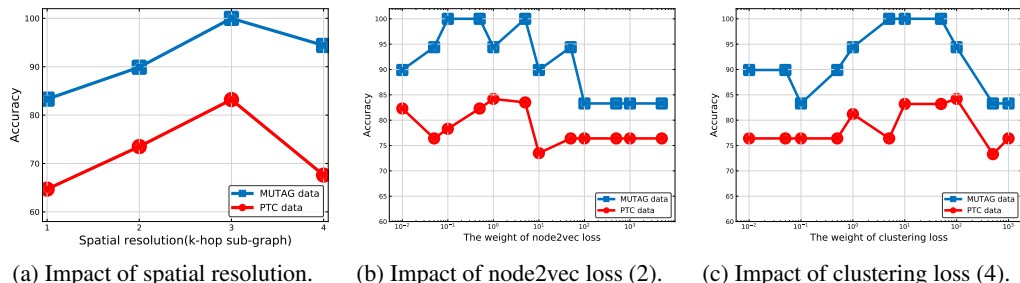

(a) Impact of spatial resolution.    (b) Impact of node2vec loss (2).    (c) Impact of clustering loss (4).

Figure 7: Impact of spatial resolution and the unsupervised losses on the prediction result.

**Impact of Spatial Resolution and Unsupervised Loss**. Impacts of spatial resolution and unsupervised loss are shown in Figure 7. In Figure 7(a), 3-hop sub-graphs tend to be a good choice for spatial resolution in bioifnormatics data. This could be consistent with the scale of meaningful functional modules in a molecule. In Figure 7(b), and (c), both the node2vec loss (2) and the clustering loss (4) are shown to benefit prediction with properly chosen weight. As expected, the node2vec loss promotes smoothness of embedding with regard to geometric interactions between sub-structures, while the clustering loss tends to promotes more "frequent" sub-structure representatives as the landmarks, thus making things more stable. Overall, the unsupervised loss serves as a regularization to benefit the learning task, and we will perform more in-depth studies in future research.

## 6 CONCLUSION

Graph neural networks provide a popular state-of-the-art computational architecture for graph mining. In this paper, we designed the SLIM network that employs structural landmarking to resolve resolution dilemmas in graph classification and capture inherent interactions in graph-structured systems.

Encouraged by the promising experimental results, we expect this attempt to open up possibilities in designing GNNs with informative structural priors.

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

## A    REPRESENTATION OF SUBSTRUCTURES

As we have discussed in Section 3.1, the simplest form for quantifying a sub-graph is

$$\mathbf{Z}_i = \mathbf{A}_i \mathbf{X}_i \tag{8}$$

Here we list a few simple variations used for representing the sub-graph that is grown from each node, including (1) emphasize the center node,

$$\mathbf{Z}_i = [\mathbf{X}_i;\ \mathbf{A}_i \mathbf{X}_i] \tag{9}$$

as inspired by (Xu et al., 2019); (2) layer-wise node type distribution

$$\mathbf{Z}_i = [\tilde{\mathbf{A}}_i^{(1)}\mathbf{X}_i; \ \tilde{\mathbf{A}}_i^{(2)}\mathbf{X}_i; \ ... \ \tilde{\mathbf{A}}_i^{(k)}\mathbf{X}_i] \tag{10}$$

, where $\tilde{\mathbf{A}}_i^{(k)}$ specifies whether two nodes in $\mathcal{G}_i$ are *exactly* $k$-hops away; or (3) weighted Layer-wise summation

$$\mathbf{Z}_i = \alpha_k \sum_k \tilde{\mathbf{A}}_i^{(k)}\mathbf{X}_i \tag{11}$$

where $\alpha_k$'s are non-negative weighting that decays with $k$, which is a more delicate summary of node-type distributions on each layer of BFS.

## B  MUTUAL COHERENCE OF STRUCTURAL LANDMARKS

In the following, we quantify a lower-bound of the coherence as a factor of the landmark size $K$ in clustering-based basis selection, since the sparse coding and $k$-means algorithm generate very similar code vectors (Coates & Ng, 1993).

**Theorem 1.** *The lower bound of the squared mutual coherence of the landmark vectors increases monotonically with $K$, the number of landmarks in clustering-based sparse dictionary learning.*

$$\mu^2(\mathbf{U}) \ \geq \ 1 - \frac{4C_d C_p}{u_{max}^2 K^{\frac{1}{d}}} \left( \left\lfloor \left(\frac{K}{2}\right)^{\frac{1}{d}} \right\rfloor^{-1} + 1 \right)$$

*Here, $d$ is the dimension, $C_d = \frac{3}{2}\left(1 + \log(d)/d\right)\gamma_d V_d$, where $\gamma_d = 1 + d\log(d\log(d))$ and $V_d = 2\Gamma(\frac{1}{2})^d/d\Gamma(\frac{d}{2})$ is the volume of the $d$-dimensional unit ball; $u_{max}$ is the maximum $\ell_2$-norm of (a subset) of the landmark vectors $\boldsymbol{\mu}_k$'s, and $C_p$ is a factor depending on data distribution $p(\cdot)$.*

*Proof.* Suppose we have $n$ spatial instances embedded in the $d$-dimensional latent space as $\{\mathbf{z}_1, \mathbf{z}_2, ..., \mathbf{z}_n\}$, and the landmarks (or codevectors) are defined as $\boldsymbol{\mu}_1, \boldsymbol{\mu}_2, ..., \boldsymbol{\mu}_K$. Let $p(\mathbf{z})$ be the density function of the instances. Define the averaged distance between the instance and the closest landmark point as

$$s = \frac{1}{n}\sum_{i=1}^n \|\mathbf{z}_i - \boldsymbol{\mu}_{c(i)}\|_2, \tag{12}$$

where $c(i)$ is the index of the closest landmark to instance $i$. As expected, $s$ will decay with the number of landmarks with the following rate (Ron Meir, 1999)

$$s \leq C_d C_p \left( \left\lfloor \left(\frac{K}{2}\right)^{\frac{1}{d}} \right\rfloor^{-1} + 1 \right) K^{-\frac{1}{d}} \tag{13}$$

where $C_d$ is a dimension-dependent factor $C_d = \frac{3}{2}\left(1 + \frac{\log(d)}{d}\right)\gamma_d$, with $V_d = 2\Gamma(\frac{1}{2})^d/d\Gamma(\frac{d}{2})$ the volume of the unit ball in k-dimensional Euclidean space and $\gamma_d = 1 + d\log(d\log(d))$; $C_p = \left(\int p(\mathbf{z})^{\frac{d}{d+1}} d\mathbf{z}\right)^{\frac{d+1}{d}}$ is a factor depending on the distribution $p$.

Since $s$ is the average distortion error, we can make sure that there exists a non-empty subset of instances $\Omega_z$ such that $\|\mathbf{z}_i - \boldsymbol{\mu}_{c(i)}\| \leq s$ for $i \in \Omega_z$. Next we will only consider this subset of instances and the relevant set of landmarks will be denoted by $\Omega_u$. For the landmarks $\boldsymbol{\mu}_p \in \Omega_u$, we make a realistic assumption that there are enough instances so that we can always find one instance $\mathbf{z}$ falling in the middle of $\boldsymbol{\mu}_p$ and its closest landmark neighbor $\boldsymbol{\mu}_p$. In this case, we have then bound the distance between the closest landmark pairs as

$$\|\boldsymbol{\mu}_p - \boldsymbol{\mu}_q\| \leq \|\boldsymbol{\mu}_p - \mathbf{z}\|_2 + \|\boldsymbol{\mu}_q - \mathbf{z}\|_2 \leq 2s.$$

For any such pair, assume that the angle spanned by them is $\theta_{pq}$. we can bound the angle between the two landmark vectors by

$$\sin(\theta_{pq}) \leq \frac{2s}{\|\boldsymbol{\mu}_p\|}. \tag{14}$$

Let $u_{max} = \max_{\boldsymbol{\mu}_p \in \Omega_u} \|\boldsymbol{\mu}_p\|_2$, we can finally low-bound the normalized correlation between close landmark pairs, and henceforth the coherence of the landmarks, as

$$
\begin{aligned}
\mu^2(\mathbf{U}) &\geq \max_{p,q \in \Omega_u} \cos^2(\theta_{pq}) \\
&= \max_{p,q \in \Omega_u} 1 - \sin^2(\theta_{pq})^2 \\
&\geq 1 - \frac{4s^2}{u_{max}^2} \\
&\geq 1 - \frac{4C_d K^{-\frac{1}{d}}}{u_{max}^2} \left( \left\lfloor \left( \frac{K}{2} \right)^{\frac{1}{d}} \right\rfloor^{-1} + 1 \right)
\end{aligned}
$$

This indicates that the squared mutual coherence of the landmarks has a lower bound that consistently increases when the number of the landmark vectors, $K$, increases in a dictionary learning process. $\quad\square$

This theorem provides important guidance on the choice of structural resolution. It shows that when a clustering-based dictionary learning scheme is used to determine the structural landmarks, the size of the dictionary $K$ can not be chosen too large; or else the risk of overfitting can be huge. Note that exact sub-structure matching as is often practiced in current graph mining tasks corresponds to an extreme case where the number of landmarks, $K$, equals the number of unique sub-structures; therefore it should be avoided in practice. The structural landmarking scheme is a flexible framework to tune the number of landmarks, and to avoid overfitting.

## C  CHOICE OF SPATIAL AND STRUCTURAL RESOLUTIONS

The spatial resolution determines the "size" of the local sub-structure (or sub-graph), such as functional modules in a molecule. Small sub-structures can be very limited in terms of their representation power, while too large sub-structures can mask the right scale of the local components crucial to the learning task. An optimal spatial resolution can be data-dependent. In practice, we will restrict the size of the local sub-graphs to 3-hop BFS neighbors, considering that the "radius" of the graphs in the benchmark data-sets are usually around 5-8. We then further fine-tune the spatial resolution by assigning a non-negative weighting on the nodes residing on different layers from the central code in the local subgraph. Such weighting is shared across all the sub-graphs and can be used to adjust the importance of each layer of the BFS-based sub-graph. The weighting can be chosen as a monotonously decaying function, or optimized through learning.

The choice of structural resolution has a similar flavor in that too small or too large resolutions are neither desirable. On the other hand, it can be adjusted conveniently by tuning the landmark set size $K$ based on the validation data. In our experiments, $K$ can be chosen by cross validation; for simplicity, we fix $K = 100$.

Finally, note that geometrically larger substructures (or sub-graphs) are characterized by higher variations among instances due to the exponential amount of configuration. Therefore, the structural resolution should also commensurate with spatial resolutions. For example, substructures constructed by 1-hop-BFS may use a smaller landmark size $K$ than those with 3-hop-BFS. In our experiments we do not consider such dependencies yet, but will study it in our future research.

## D  HIERARCHICAL VERSION

### D.1  SUBTLETY IN SPATIAL RESOLUTION DEFINITION

First we would like to clarify a subtlety in the definition of spatial resolutions. In physics, resolution is defined as the smallest distance (or interval) between two objects that can be separated; therefore it must involve two scales: the scale of the object, and the scale of the interval. Usually these two scales are proportional. In other words, you cannot have a large intervals and small objects, or the opposite (a small interval and large object). For example, in the context of imaging, each object is a pixel and the size of the pixel is the same as the interval between two adjacent pixels.

In the context of graphs, each object is a sub-graph centered around one node, whose scale is manually determined by the order of the BFS-search centered around that node. Therefore, the interval between two sub-graphs may be smaller than the size of the sub-graph. For example, two nodes $i$ and $j$ are direct neighbors, and each of them has a 3-hop sub-graph. Then, the interval between these two subgraphs, if defined by the distance between $i$ and $j$, will be 1-hop; this is smaller than the size of the two sub-graphs, which is 3-hop. In other words, the two objects/subgraphs indeed overlap with each other, and the scale of the object and the scale of the interval between objects is no longer commensurate (large objects and small interval in this scenario).

This scenario makes it less complete to define spatial resolutions just based on the size of the sub-graphs (as in the main text), since there are actually two scales to define. To avoid unnecessary confusions, we skip these details. In practice, one has two choices dealing with the discrepancy: (1) requiring that the sub-graphs are not overlapping, i.e., we do not have to grow one $k$-hop subgraph around each node; instead, we just explore a subset of the sub-graphs. This can be implemented in a hierarchical version which we discuss in the next subsection; (2) we still allow each node to have a local sub-graph and study them together, which helps cover the diversity of subgraphs since theoretically, an ideal choice of the subgraph is highly domain specific and having more sub-graph examples gives a better chance to include those sub-graphs that are beneficial to the prediction task.

### D.2 HIERARCHICAL SLIM

We can implement a hierarchical version of SLIM so that sub-graphs of different scales, together with the interacting relation between sub-graphs under each scale, can be captured for final prediction. Note that in (Ying et al., 2018) a hierarchical clustering scheme is used to partition one graph, in a bottom up manner, to less and less clusters. We can implement the same idea and construct a hierarchy of scales each of which will host a number of sub-structures. The structural landmarking scheme will be implemented in each layer of the hierarchy to generate graph-level features specific to that scale. Finally these features can be combined together for graph classification.

## E SEMI-SUPERVISED SLIM NETWORK

The SLIM network is flexible and can be trained in both fully supervised setting and semi-supervised setting. This is because the SLIM model takes a parametric form and so it is inductive and can generalize to any new samples; on the other hands, the clustering-based loss term in (4) can be evaluated on both labeled samples and unlabeled samples, rendering the extra flexibility to look into the distribution of the testing sample in the training phase, if they are available. This is in flavor very similar to the smoothness constraint widely used in semi-supervised learning, such as the graph-regularized manifold learning (Belkin et al., 2006). Therefore, the SLIM network can be implemented in the following modes

- Supervised version. Only training graphs and their labels are available during the training phase, and the loss function (4) is only computed on the training samples.

- Semi-supervised version. Both labeled training graphs and unlabeled testing graphs are available. The loss function (4) will be computed on both the training and testing graphs, wile the classification loss function will only be evaluated on the training graph labels.

## F INTERPRETABILITY

The SLIM network not only generates accurate prediction in graph classification problems, but can also provide important clues on interpreting the prediction results, because the graph-level features in SLIM bear clear physical meaning. For example, assume that we use the interaction matrix $\mathbf{C}_i$ for the $i^{th}$ graph $\mathcal{G}_i$ as its feature representation; and the $pq^{th}$ entry then quantifies the connectivity strength between the $p^{th}$ sub-structure landmark and the $q^{th}$ structure landmark. Then, by checking the $K^2$-dimensional model coefficients from the fully-connected layer, one can then tell which subset of substructure-connectivity (i.e., two substructures are directly connected in a graph) is important in making the prediction. To improve the interpretability one can further imposes a sparsity constraint on the model coefficient.

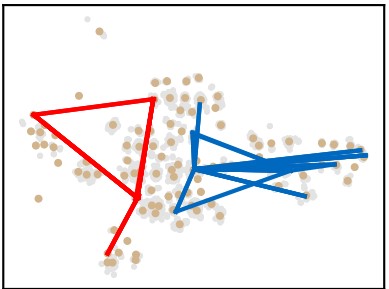 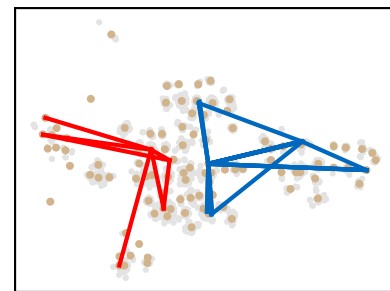

Figure 8: Two pairs of molecules from different classes of MUTAG data, with their respective substructure-interaction-patterns marked on top of the learned structural landmarks. Red and blue colors signify the class labels.

In traditional graph neural networks such as GraphSAGE of GIN, node features are transformed through many layers and finally mingled altogether through graph pooling. The resultant graph-level representation, whose dimension is manually determined and each entry pools the values across all the nodes in the graph, can be difficult to interpret.

In Figure 8, we illustrate the sub-structure embedding and the learned landmarks of SLIM network on the MUTAG dataset. Here, molecules belonging to different classes are marked with red and blue edges. As can be observed, the sub-structures landmarks can generate highly discriminative features for graph classification; furthermore, by examining the substructure instances associated with each landmark, domain experts can acquire valuable clues for interpretating the underlying mechanism of classification. In fact, discriminative roles of the landmarks are two-fold. First, landmark themselves can be discriminative by being associated with one class more often than the other. Second, even in scenarios where both classes may share some common sub-structure landmarks (which is possible in molecules due to some common functional modules), the interacting pattern defined among the landmarks can still serve as an effective discriminator. We believe the dual discriminating mechanism using structural landmarks can be quite desirable in solving difficult graph classification problem, which we will investigate in more detail in our future studies.

## G    ABLATION STUDY (TEST ACCURACY EVOLUTION)

### G.1    ALGORITHM STABILITY

Here we have reported the evolution of the testing accuracies for a number of competing algorithms on more all the benchmark training data, as in Figure 9.

### G.2    TIME COMPLEXITY

We also report the relationship between the accuracy and time of SLIM and several other methods to complete 200 epochs on the MUTAG and PTC data sets. As in Figure 10 The accuracy of our method is better than most methods on the two data sets.When running for 200 epochs, our method is faster than most methods on the MUTAG data set.  The time cost of our approach is mainly on the calculation of spatial resolution, structural resolution and back propagation when updating model parameters. Assuming that the size of each mini-batch size is $n$, the average number of nodes in each graph is $m$, and each batch is running for all nodes in the graph to establish the computational spatial resolution required $O(nm \cdot |\overline{\mathcal{N}_k}|)$ time, where $|\overline{\mathcal{N}_k}|$ is the averaged $k$-hop-neighbor size. Considering that DEC clustering is needed to calculate the structural resolution, this part needs to minimize the KL divergence between it and the auxiliary target distribution, assuming $k_0$ is the number of landmarks, the time complexity is only $O(nm \cdot k_0)$ .

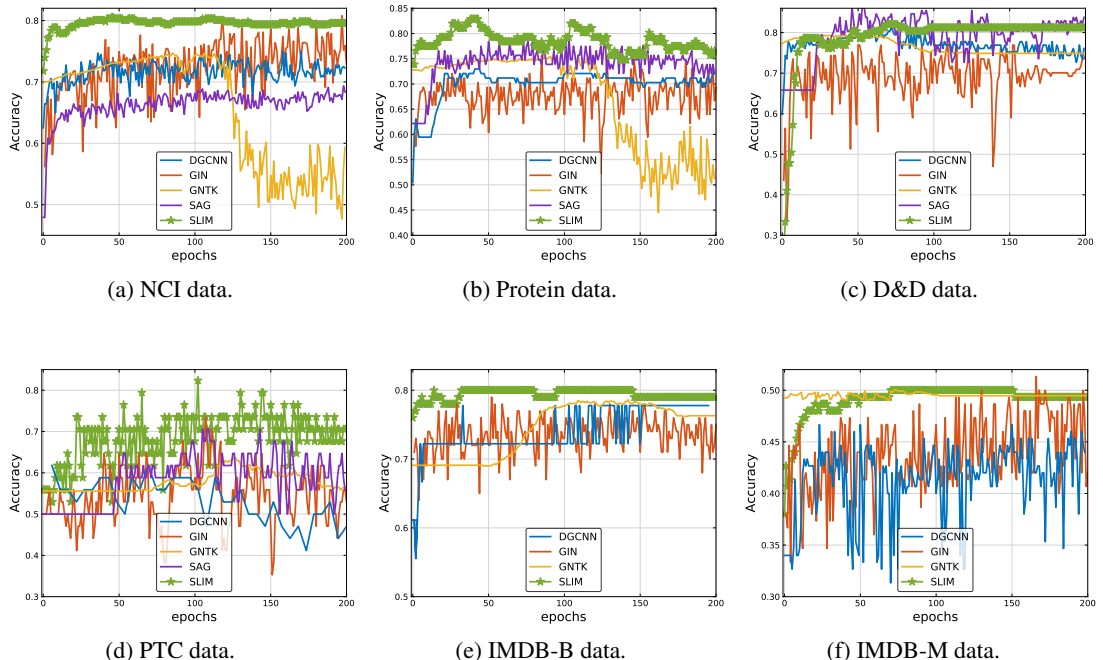

Figure 9: Testing accuracy of different algorithms over the training epochs.

Table 4: Performance of SLIM using different final graph features.

| feature | MUTAG | PTC | NCI1 | Protein | D&D | IMDB-B | IMDB-M | COLLAB |
|---|---|---|---|---|---|---|---|---|
| SLIM-GNN | 92.11±2.27 | 72.32±4.16 | 80.44±2.03 | 77.21±1.49 | **79.61±2.66** | 78.40±1.12 | 53.17±3.19 | **78.22±2.02** |
| SLIM-Concat | 90.27±4.12 | 70.28±7.25 | 79.21±3.18 | 75.35±3.27 | 79.04±3.07 | **80.20±2.10** | **54.00±4.02** | 77.61±0.77 |
| SLIM-C | **93.28±3.36** | **74.41±6.92** | **80.53±2.01** | **77.47±2.01** | 78.33±5.74 | 79.80±4.21 | 53.20±4.36 | 76.58±1.45 |

## H  RICH GRAPH LEVEL FEATURES

The structural landmarking mechanism allows computing rich graph-level features by using different approaches to project structural details of each graph onto the common space of landmarks.

- SLIM-C. Using the (normalized) sub-structure interaction matrix for each graph as its feature, as discussed in Section 3.3.

- SLIM-Concat. Concatenating the density, mean, and interaction features discussed in Section 3.3 together (after re-shaping into vectors). One could also transform the interaction feature to a smaller matrix via bilateral dimension reduction before reshaped into a vector. Then a fully connected layer follows for the final prediction.

- SLIM-GNN (Landmark Graph). Each graph $\mathcal{G}_i$ can be transformed into a landmark-graph a with fixed number of $K$ (landmark) nodes, with $\mathbf{p}_i$ and $\mathbf{C}_i$ quantifying the weight of each node and the edge between every pair of nodes, and $\mathbf{M}_i$ the feature of each node (see definition in Section3.3). Then, this graph can be subject to a graph convolution such as $\mathbf{D}_i^{-1}\mathbf{C}_i\mathbf{M}_i$ generate a fixed-dimensional graph-level feature without having to take care of the varying graph size. We will study this in our future experiments.  In Table 4, we compare the performance of using different features of the projected graph. Overall, interaction matrix $\mathbf{C}_i = \mathbf{W}_i \cdot \mathbf{A}_i \cdot \mathbf{W}_i'$, which encodes the interacting relations (geometric connections) among the $K$ structural landmarks are slightly better than GNN-features and concatenation features, except that SLIM-GNN have lower fluctuations.For most bioinformatics data, the SLIM-C gains a competitive score on MUTAG, PTC, NCI1 and Protein data sets.

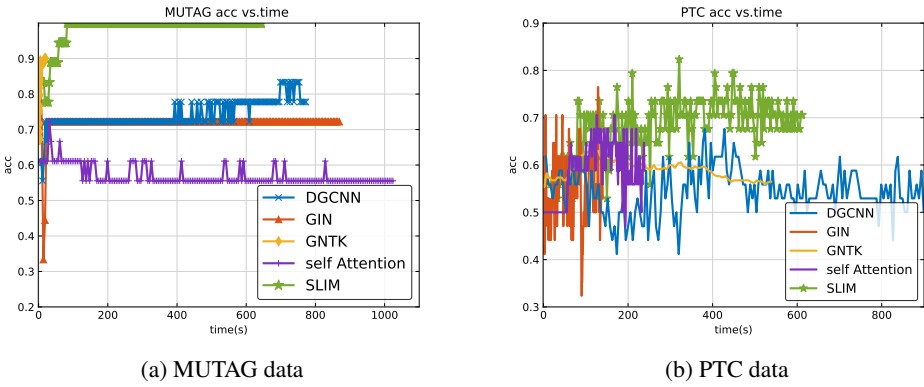

(a) MUTAG data            (b) PTC data

Figure 10: Average wall-clock time comparisons of SLIM and other methods.

## I INTERACTION VERSUS INTEGRATION

The SLIM network and existing GNNs represent two different flavors of learning, namely, interaction modelling versus integration approach. Interaction modelling is based on mature understanding of complex systems and can provide physically meaningful interpretations or support for graph classification; integration based approaches bypass the difficulty of preserving the identity of sub-structures and instead focus on whether the integrated representation is an injective mapping, as typically studied in graph isomorphism testing.

Note that an ideal classification is different from isomorphism testing and is not injective. In a good classifier, the goal of deciding which samples are similar and which are not are equally important. Here comes the tradeoff between handling similarity and distinctness. The Isomorphism-flavor GNN's are aimed at preserving the differences between local sub-structures (even just a very minute difference), and then map the resultant embedding to the class labels. Our approach, on the other hand, tries to absorb patterns that are sufficiently close to the same landmark, and then map the landmark-based features to class labels. In the latter case, the structural resolution can be tuned in a flexible way to explore different fineness levels, thus tuning the balance between "similarity" and "distinctness"; in the meantime, the structural landmarks allow preserving sub-structure identities and exploiting their interactions.

## J COMPARISON WITH RELATED METHODS

Graph kernels are powerful methods to measure the similarity between graphs. The key idea is to compare the sub-structure pairs from the two graphs and compute the accumulated similarity, where examples of substructures include random walks, paths, sub-graphs, or sub-trees. Among them, paths/sub-graphs/sub-trees are deterministic sub-structures in a graph, while random walks are stochastic sequences (of nodes) in a graph.

Although the SLIM network considers sub-structures as basic processing unit, it has a number of important differences. First, we consider sub-structure landmarks that are **end-to-end optimizable**, aimed at both reconstructing substructure distribution and generating discriminative interacting pattern for graph classification; while sub-structures in graph kernels or graphlets are typically identified offline. Second, graph kernels measure similarity between all possible pairs of sub-structures across two graphs; while SLIM network models interaction between sub-structures, i.e., the functional organization of a graph, which is very different from graph kernels. Third, interpretation is challenging due to nonlinearity of kernel methods and exponential number of candidates in graphlets; while SLIM maintains a reasonable amount of discriminative "landmark" sub-structures easier to interpret.

In recent years, embedding algorithms have drawn considerable attention that transform nodes (or sub-graphs) into a low-dimensional Euclidian space, as pioneered by the word-to-vector work (Mikolov et al., 2013). It is worthwhile to note that these algorithms focus on node- or edge-level embedding,

while our target is graph-level classification. As a result, our approach emphasizes more on the innovation in modelling the interacting relation between component parts of a graph, as inspired by the views from complex systems. In fact, our most important contribution lies exactly in learning substructure landmarks jointly across graphs to enable identity-preserving graph pooling. This is rarely a consideration in algorithms whose main focus is just to embed the nodes or sub-graphs.

Here we use Role2Vec (Ahmed et al., 2018) as an example to illustrate the similarity and difference between our approach and embedding-type methods. Similarity: both methods embed nodes or subgraphs, and consider high-order subgraph features. Differences: (1) Tasks are different; Role2Vec is node-level embedding, while ours is graph-level classification; (2) Data are different; Role2Vec focuses on single graph, we simultaneously handle many graphs, i.e., align sub-structures from different graphs and project each graph onto shared structural landmarks - a new framework for graph pooling; (3) Methods are different; Role2Vec designs attributed random walk, we use KL to fit substructure distribution; R2V finds subgraph motifs offline like in graphlets, we optimize discriminative substructure landmarks in an end-to-end fashion.

Recent years, various hierarchical pooling strategies (Ying et al., 2018; Lee et al., 2019; Khasahmadi et al., 2020) have been proposed to fully exploit non-flat graph organization and show promising results in graph classification. However, despite the hierarchical process that reduces the number of nodes layer by layer, the final representation is still in the form of a single, aggregated node vector for feature compatibility, leading to potential loss of informative graph structural details. Note that hierarchical methods usually perform grouping inside each graph, while in our approach the substructure landmarks are identified by jointly clustering the substructure instances across all the graphs. Therefore our landmark set size is typically larger than the number of clusters in hierarchical methods, in order to accommodate the diversity in sub-structure distribution. Note that some of the hierarchical methods need to sort the nodes of the graph as a pre-processing procedure (Ying et al., 2018; Khasahmadi et al., 2020).

We are also aware some other types of aggregation strategies: Sortpooling re-arranges graph nodes in a linear chain and perform 1d-convolution (Zhang et al., 2018); SEED uses distribution of multiple random walks to capture graph structures (Wang et al., 2019); Deep graph kernel evaluates graph similarity by subgraph counts (Yanardag & Vishwanathan, 2015). Again, explicit modelling of the interaction between constituent parts of the graph is not considered in these approaches.

