# OpenReview forum: "Structural Landmarking and Interaction Modelling: on Resolution Dilemmas in Graph Classification"
_ICLR.cc/2021/Conference — Reject_

### Official Review · AnonReviewer4 · 2020-10-27
**Review comments to Paper 1160**

**Rating:** 6
**Confidence:** 4

**Review:**

==========Summary==========

In this paper, the authors investigate how to learn graph representations from structural landmarks. In particular, SLIM, an end-to-end trainable method, is proposed to simultaneously learn landmark and graph representations guided by downstream graph classification signals. Empirical results demonstrate the effectiveness of the proposed method on public benchmark datasets.

==========Reason for the rating==========

At this moment, I am standing between 4 and 5. Overall, I feel the technique proposed in this paper is potentially interesting. However, its current presentation could also deliver quite confusing information. Hopefully, the authors could address my concerns in the rebuttal.

==========Strong points==========
1. Structural landmark is an interesting idea to approach graph representation learning. This study goes beyond the assumption of "bag of structures", and investigates the angle of modeling interaction between discovered structures.

2. The authors propose the SLIM framework that implements the proposed idea well.

3. The provided empirical evidences suggest the SLIM technique is promising in graph classification tasks.

==========Weak points==========
1. The key insight "resolution dilemma" could be confusing. While one expects graph pooling causes the struggling between spatial and structural resolution, the two resolutions seem to be two orthogonal dimensions.
    - The discussion on spatial resolution seems to be around the assumption for graph classification: bag of structures (BOS) or some more complex model that considers relations between discovered structures. Existing graph pooling methods mostly assume the BOS model, and develop different "counting methods" to generate one fixed-size vector for each graph. From the perspective of counting, it is difficult to see how existing pooling reduces spatial resolution, while indeed such pooling methods cannot consider relations between structures. Could the authors clarify what exactly "diminish" means in the context of "spatial resolution"?
    - The discussion on structural resolution seems to be around model selection in terms of expressive power from learned representations: if we really can make a choice, which one could result in the most promising generalization performance. This problem is indeed meaningful, but it is not caused by pooling layers and is not necessary to address it at pooling layers.
    - Overall, it is difficult to see the connection between section 2 and 3. After zooming into the two dimensions in section 2, they seem orthogonal, with weak coupling with pooling layers. The authors may need to provide stronger reasoning or evidences to motivate the proposed technique from the perspective of "resolution dilemma".

2. The expressive power of substructure embedding could be a potential concern. Using $A^{(k)}$, different layouts within k-hops (k > 1) may not be able to be differentiated anymore. Does this imply different structures within k-hops do not matter?

3. The presentation in section 3 could be further improved.
    - The details on how to end-to-end train SLIM is missing in section 3, while one may find some relevant information in the experiment section. To make each section self-contained and the technique clearly described, the authors may put down the relevant contents in section 3.
    - Section 3.3 is definitely critical in the whole framework; however, the current presentation could unnecessarily increase the difficulty in understanding the idea behind.

==========Questions during rebuttal period==========

Please address and clarify the weak points above.

In addition, the current draft could use more proofreading to clear typos and grammar errors. The idea of landmark has also been discussed in a recent work [1], in the context of unsupervised learning. The authors may need to highlight their unique perspective.

==========Reference==========
[1] DDGK: Learning Graph Representations for Deep Divergence Graph Kernels, WWW 2019

==========Post rebuttal comments==========

The rebuttal has addressed my main concerns. In general, I believe the core idea of modeling substructure interactions in GNNs should be shared in front of more audience. Therefore, I have increased the rating accordingly.

---

> ### Author Response · Authors · 2020-11-15
> **We highly appreciate the valuable comments and we have followed the comments in improving our paper.**
>
> We highly appreciate the valuable comments from the reviewer. Indeed, these comments help deepen our own understanding, and allow us to further improve/clarify the presentation. Our detailed responses and revisions are summarized below.
>
> $Q$: While one expects graph pooling causes the struggling between spatial and structural resolution, the two resolutions seem to be two orthogonal dimensions ... The authors may need to provide stronger reasoning … from the perspective "resolution dilemma".
>
> $A$: We highly appreciate this in-depth discussion. We are sorry for the confusion and following is our clarification.
>
>     We have two dilemmas: one for spatial resolution selection, and one for structural resolution selection. The two resolutions are not fighting directly. In fact, the two dilemmas have a causal relation, as discussed in "Relation between spatial and structural resolution dilemmas" (previously the last paragraph of Sec 2, now Sec 2.3). The logic is  ("->" means "leads to"):
>
>     Exact sub-graph matching -> [structural resolution explodes/dilemma] -> problematic to compare sub-structure across graphs -> have to use collapsing-style graph pooling -> [spatial resolution diminishes/dilemma]
>
>     That is, a bad (exploding) structural resolution leads to a bad (diminishing) spatial resolution. Now, SLIM replaces exact matching with landmarking, so the structural resolution becomes well controlled. The logic is updated as:
>
>     Structural landmarking -> [structural resolution controlled (dilemma solved)] -> easy to compare sub-structures across graphs -> identity-preserving graph pooling applicable-> [spatial resolution improved (dilemma solved)]
>
>     Namely, both resolutions are better selected, and both dilemmas are solved. This seems less of a struggle (a zero-sum game), or orthogonal (two act independently). We could see it as a coordinated, win-win solution by our method.  Full discussions can be found in Section 2.3.
>
>
> $Q$: Could the authors clarify what exactly "diminish" means in the context of spatial resolution?
>
> $A$: Resolution is the ability to identify fine details/parts. After a graph is pooled/squeezed into a single vector, we can no longer differentiate its parts, i.e., nodes or subgraphs. We can only see the graph as a whole, which is the lowest resolution possible so we call it "diminishing".  (If one can identify the finest parts, i.e., each node, then it will be the highest spatial resolution).
>
> $Q$: The discussion on structural resolution … is indeed meaningful, but it is not caused by pooling layers and is not necessary to address it at pooling layers.
>
> $A$: The reviewer is correct. Based on the logic chain above: (collapsing-style) pooling does not cause bad structural resolution; it is the bad, exploding structural resolution that makes collapsing-style graph pooling an inevitable choice. Now SLIM chooses a better (non-exploding) structural resolution, and so identity-preserving graph pooling becomes possible.
>
> $Q$: The expressive power of substructure embedding … A(k), may not be able to be differentiated anymore
>
> $A$: $A^{(k)}$ can quantify sub-structure differences to a great extent because it looks into node distribution on different "layers" of each subgraph from the center node. It may not be strictly injective as GIN layer (a choice suggested in Sec 3.1), but it serves as a simple block for embedding and works well.
>
> $Q$: Section 3 could be further improved. The details on how to end-to-end train SLIM is missing in section 3 … the authors may put down the relevant contents in section 3.
>
> $A$: We thank the reviewer for this suggestion. We have revised Section 3. In particular, we have added Fig.4 to illustrate details of the end-to-end training architecture (some parts of it moved from experimental section as reviewer suggested). In Fig.4, we have clearly marked the three main steps of our method (green shades), their variables, and related equations. We have also marked the loss function, and specified network parameters in related discussion (end of Section 3). We believe the new version clarifies a lot.
>
>
> $Q$: Section 3.3 is definitely critical in the whole framework; however, the current presentation could unnecessarily increase the difficulty in understanding the idea behind.
>
> $A$: We have revised Section 3.3; we believe now it’s clearer and allows readers to easily implement the idea.
>
>
> $Q$: The current draft could use more proofreading to clear typos
>
> $A$: We have thoroughly proofread the paper and made corrections.
>
>
> $Q$: The idea of landmark has also been discussed in a recent work [1] … The authors may need to highlight their unique perspective.
>
> $A$: We thank the reviewer for pointing out Deep Divergence Graph Kernel (DDGK), an interesting work that applies the landmark idea with new attention mechanisms for unsupervised graph similarity learning. We have included it in our reference, as well as the discussion in Section 4.2.

---

> > ### Comment · AnonReviewer4 · 2020-11-24
> > **Response to the Authors**
> >
> > Hi authors,
> >
> > Appreciate your hard work on addressing my questions. My concerns have been cleared. I have increased the rating accordingly.

---

> > > ### Author Response · Authors · 2020-11-24
> > > **We are glad to address reviewer's concerns**
> > >
> > > We are glad to have addressed the reviewer's concerns, and we appreciate your valuable comments.

---

> ### Author Response · Authors · 2020-11-24
> **We hope our responses have addressed the concerns of the reviewer.**
>
> We hope our responses have addressed the concerns of the reviewer.
>
> In case you have any further concern or question, please let us know and we will be glad to take them and make revisions or clarifications.

---

### Official Review · AnonReviewer1 · 2020-10-28

**Rating:** 6
**Confidence:** 4

**Review:**

Strengths

1. This paper studied an important problem of graph mining on graph classification tasks by investigating the challenges that previous graph models encountered and solving them with the proposed method.

2. The developed method is novel and interesting by proposing an inspiring concept called structural landmarking and capturing inherent interactions with it.

3. The analysis and survey of related work with respect to the two mentioned resolution types is comprehensive.

4. The paper is well written and easy to follow.

Weakness

1. The paper lacks enough analyses of behaviors of learned structural landmarks, with only an analysis of choices of their numbers.

2. The time complexity of the developed method is not analyzed and its running time comparison with other baseline methods are also missing.

3. The analysis of how the choices of each part of features learned in graph pooling would affect the results is missing.

4. The experimental part lacks the analysis of why the method performs very well on some datasets while not performing well on others.

Summary: This paper studies the problem of graph classification on chemical and social datasets. Existing graph classification methods with graph neural networks learn node embeddings via aggregation of neighbors and combine all node features into a final graph feature for classification, while such operations usually lack the ability for identifying and modelling the inner interactions of substructures. To remedy the information loss in graph pooling, the authors leverage the learned substructure landmarks to project graphs onto them for modelling the interacting relations between component parts of a graph. In this regard, an inductive neural network model for structural landmarking and interaction modelling is developed to resolve potential resolution dilemmas in graph classification and capture inherent interactions in graph-structured systems. Empirical experiments on both chemical and social datasets validate the effectiveness of the method. Generally speaking, the paper is well written and easy to follow, with clear motivation and organization. However, I have concerns about the lack of analysis for learned structural landmarks, since in the paper only the choice of its number is well discussed. Also, the time complexity of the developed method is not well studied. The detailed comments and suggestions of this paper are as follows.

1.The paper proposes to learn structural landmarks and obtain representations of graphs by projecting them. Therefore, the quality of learned landmarks is crucial while the paper lacks enough analyses for them. I suggest providing a more comprehensive analysis for them.

2.The proposed method generates various kinds of graph-level features while lacking enough analyses of their impacts on results. I suggest conducting more experiments of ablation study for this part.

3.The detailed statistics of benchmark datasets are not mentioned such as the distribution of number of nodes in graphs.

4.Although the results are competitive compared with other baselines, the authors didn’t explain why the method performs well on some datasets while not performing well on others. I suggest analyzing the reasons comprehensively.

5.Only one evaluation metric is used in experiments, which is the accuracy. Since it’s a classification task, I suggest using various metrics to show the effectiveness.

6.There are some typos in the paper that requires double checking: For example, "breath-first search " -> "breadth-first search", "an molecule" -> "a molecule"

---

> ### Author Response · Authors · 2020-11-15
> **We highly appreciate the valuable comments and we have followed the comments in improving our paper.**
>
> We highly appreciate the reviewer for the constructive and encouraging comments. We have followed all the suggestions of the reviewer, which greatly improves the paper. Our response is summarized as follows.
>
> $\bf{Q}$: The paper proposes to learn structural landmarks and obtain representations of graphs by projecting them. Therefore, the quality of learned landmarks is crucial while the paper lacks enough analyses for them. I suggest providing a more comprehensive analysis for them.
>
> $A$: The reviewer is correct that the landmarks are of crucial importance to the final prediction. Therefore, we have been very careful in designing the loss function that promotes the learning of useful landmarks, as discussed in detail in Section 3.2 (1st paragraph). That is, the landmarks should have high statistical coverage or be somewhat “frequent” (unsupervised loss), and should lead to discriminative interaction patterns (supervised loss). We have also added new ablation studies in Fig. 7(b) and (c) to study the impact of the unsupervised loss on the quality of the landmarks (in terms of classification accuracy). In future studies, we will provide more concrete examples of the learned landmark structures and their implications in the context of molecule/protein classification.
>
>
> $Q$: The proposed method generates various kinds of graph-level features while lacking enough analyses of their impacts on results. I suggest conducting more experiments of ablation study for this part.
>
> $A$: As suggested by the reviewer, we have conducted more ablation studies on different types of final graph features, and included them in the Table 4 in the Appendix.  As can be seen, the "SLIM-concatenation" feature and "SLIM-GCN"features are slightly worse than the normalized interaction features (currently reported), but they are still quite competitive against other methods. Detailed discussions are in Appendix H.
>
>
> $Q$: The detailed statistics of benchmark datasets are not mentioned such as the distribution of number of nodes in graphs.
>
> $A$: As the reviewer suggested, we have provided the detailed statistics of the benchmark graph data sets in Table 1.
>
>
> $Q$: Although the results are competitive compared with other baselines, the authors didn’t explain why the method performs well on some datasets while not performing well on others. I suggest analyzing the reasons comprehensively.
>
> $A$:  Our method has the highest accuracy on 6 of the 8 benchmark data sets, among 10 competing methods. For the remaining 2 data sets: (1) On NCI1 data, our method is the second best; in fact, it outperforms all other neural network methods and is only inferior to graph kernel method; (2) On COLLAB data, our method is worse and ranks No. 6. We speculate that COLLAB data have no node attributes, and have many densely connected subgraphs, therefore, the advantage of structural landmarking becomes less significant. We have added this discussion in Section 5.
>
>
> $Q$: Only one evaluation metric is used in experiments, which is the accuracy. Since it’s a classification task, I suggest using various metrics to show the effectiveness.
>
> $A$: We totally agree with the reviewer. We used accuracy mainly because it is a popular choice in the literature of graph classification. As the reviewer suggested, other metrics can make the comparison more comprehensive (precision, recall, F1-score, AUC curve), especially when the data are imbalanced, or there is a tradeoff between sensitivity and specificity. Hopefully, the imbalance of the benchmark data we have used seems to be less of a concern (as shown in Table 1, right-most column, six among eight data sets are exactly or roughly balanced). We will definitely investigate this in more detail and make a more comprehensive evaluation in our future studies.
>
>
> $Q$: There are some typos in the paper that requires double checking: For example ...
>
> $A$: We thank the reviewer for the detailed comments on our writing. We have made a thorough proofread and correction.

---

### Official Review · AnonReviewer2 · 2020-10-28
**Addressing dual "resolution dilemmas" by graph pooling with landmarks**

**Rating:** 6
**Confidence:** 2

**Review:**

### Summary

The proposed SLIM algorithm organizes graph neural networks around substructures surrounding "landmarks" in the graph.  In addition to presenting the three steps of the SLIM algorithm (sub-structure embedding, sub-structure landmarking, and "identity-preserving" graph pooling), the authors compare to other approaches on a graph classification problem.  A large part of the paper is also given over to a high-level discussion of "resolution dilemmas."

The idea of organizing around landmarks (and making the landmark choice a part of the optimization) is a nice one, and the new approach certainly seems to improve performance in the graph classification problem.  But I would have loved to see a more quantitative treatment of the "resolution dilemmas."  All the ideas in this space involve lossy compression of the graph, and the authors characterize this in terms of loss of spatial resolution (ability to understand the "high-level" topology of the graph) or loss of structural resolution (ability to understand the prevalence of different types of local structures or motifs).  The attempt to balance these types of resolution concerns is the main motivation for the current paper.  But the discussion of these types of resolution is purely qualitative; the only place where there is really a quantitative discussion is in the experimental section in Figure 4.  Given the amount of space given over to discussing these different types of resolution, and the connections made (however briefly) to regularization ideas, I would have loved so see more of the discussion focus on how spatial and structural resolution can be measured, and how those measures might be used to guide the selection of algorithm hyper-parameters like the number of landmarks.

There is also a repeated statement that using substructure landmarks makes the representation easier to interpret.  This seems intuitive; but there are no examples illustrating such interpretability.

### Typos

- Abstract: "iterpretable"
- Sec 2.1: "tow key blocks", "capture lager substructures"
- Sec 2.2: "Milo1, et al"
- Sec 3: "sub-sturcture embedding"
- Fig 4: "stuctutral resolution"

### Update

Thanks to the authors for their response to the reviews, and to the other reviewers for their comments as well.  It seems that we were all confused about many of the same things.

The authors have clarified some of the points raised, and I appreciate that.  I also continue to appreciate that the empirical results are promising.  At the same time, I personally remain confused about how to reason in a quantitative way -- suitable for diagnosing problems, determining hyperparameters, etc -- about the resolution tradeoffs that are described.  In other settings involving graph coarsenings, wavelets, graph Fourier transforms, etc, there is usually a more quantitative way of expressing what information is lost and retained by a given type of compression.  The discussion of coherence in the appendices says something about this, but not in a way that I would understand how to operationalize.

I also appreciate that the chemistry example in the appendix exists, but I do not understand how to visually interpret the picture.

My recommendation remains a weak accept, as I think it is likely worthwhile to put the empirical results out in the world, and the theory may follow.  But I remain wary of my own lack of understanding of the theory in a quantitatively meaningful way, and would also welcome the chance to read a future version that had some of these aspects more clearly worked out and explained.

---

> ### Author Response · Authors · 2020-11-15
> **We highly appreciate the valuable comments and we have followed the comments in improving our paper.**
>
>
> We highly appreciate the valuable and encouraging comments from the reviewer, as well as the in-depth summary of the main idea. The main concerns from the reviewer are addressed below.
>
>
> $Q$: I would have loved so see more of the discussion focus on how spatial and structural resolution can be measured, and how those measures might be used to guide the selection of algorithm hyper-parameters like the number of landmarks.”
>
> $A$: We totally agree with the reviewer that a quantitative analysis on the impact of the spatial and structural resolution will be extremely useful both in terms of theoretic understanding and empirical performance. In the meantime, admittedly, hyper-parameter selection can be always be challenging as in many learning algorithms. In the following, we have summarized our analysis in quantifying and selecting the two resolutions (Appendix C), ablation studies (newly added), as well as future theoretic direction (with a preliminary analysis currently presented).
>
> (1) Quantification of resolutions. Currently, the spatial resolution is directly controlled by k in the k-hop local sub-graph (around each node), and the structural resolution is controlled by the number of structural landmarks, K. These two numbers are directly tunable and so they can be deemed as a convenient quantifier of the two resolutions.
>
> (2) Selection of resolutions. We have shown empirically how these two choices affect the generalization performance. Our observation is that spatial resolution may depend on the size of the graph and the scale of meaningful substructures. For example, in molecules, it is typically good to select k as 2 or 3 hops, which is reflected in our newly added ablation study on the impact of the spatial resolution (Fig.7(a)). For structural resolution, we have shown in Fig.4 that neither too small nor too large K is desirable, and we typically set K = 100 in all our experiments. In the future, we will definitely perform more extensive study and also examine better ways to select the two resolutions.
>
> (3) We have added more ablation studies in Section 5 (Fig.7) to reveal the impact of the hyperparameters on the quality of learned structural landmarks, mainly in terms of the final classification accuracy (since better landmarks should lead to more discriminative graph-level features). These detailed studies shed more light on the main mechanism of our algorithm, and detailed discussions are in the last paragraph of Section 5.
>
> (4) In the future, we will be actively pursuing a more rigorous theoretic delineation of the proposed method, especially on the dependency of the generalization performance on the two resolutions. Currently, we have a very preliminary analysis on how the mutual coherence of the learned landmarks depends on the structural resolution K (Appendix B). It shows that too large K can lead to high redundancy in the basis, which leads to overfitting (Marsousi et al., 2014) and unstable dictionary learning (Donoho & Huo., 2001; Mehta & Gray, 2013). We will further extend this analysis to gain better insights in classification.
>
>
>
> $Q$: using substructure landmarks makes the representation easier to interpret… but there are no examples illustrating such interpretability.
>
> $A$: We previously showed a preliminary example on the learned structural landmarks and how they are used to represent graphs (Fig. 8, Appendix F). We will further dig into this example and provide domain-specific interpretations of the landmark in molecule classification.  This requires more chemistry knowledge and we are actively working on it.
>
>
> $Q$: Typos
>
> $A$: We have made a thorough proofread and corrected the typos. We highly appreciate the reviewer for the detailed comments.

---

### Official Review · AnonReviewer3 · 2020-10-28
**Interesting exploration of  spatial resolution and structural resolution in Graph Neural Networks**

**Rating:** 6
**Confidence:** 3

**Review:**

Summary:
Authors introduces two new concepts spatial resolution and structural resolution in regards to understanding the graph structured data which are quite interesting and enlightening. Idea about projecting graph information
into structural landmarks is intriguing. To help make stronger case, I would suggest to do proper ablation study as it is not clear how much gain is coming unsupervised learning.

Pros:
Overall I like intuition and the method about capturing the spatial resolution and structural resolution in a strategic manner. Author have some strong empirical performance especially on Protein, PTC and IMDB-M dataset.

Cons:
Authors argue that the classic graph neural networks which employ graph pooling operations are the bottleneck in identifying necessary substructures (or their interactions) for yielding high discriminative performance.  However, such statements are quite loose and need further theoretical justification given the fact graph pooling operations such as deepsets or sum-pooling are universal/injective functions in nature and thus can reflect any changes in the graph sub-structure (however their function smoothness or amount of representative power captured is entirely different issue).

It not clear why right hand side spectrum in Figure 2 will lead to lower generalization performance or over-fitting. Most of the time motifs/graphlets act as an atomic structure of a graph and their frequency distribution drives the discriminative performance.  As such identifying all such atomic structures should be helpful rather than harmful. It would be great if authors can expand on their explanation here and provide a real world example that would be more convincing to support their hypothesis.

There are certain paragraphs which are hard to read. For instance, Figure 1 lacks detail description and it is not clear what "all the nodes are mixed into one" means in the context. Also, a general suggestion would be to add more descriptive caption for each Figures in the paper.

I would suggest to provide compelling real-world examples (or do more qualitative analysis) besides the strong empirical performance in the main paper (there is some discussion in appendix but highlights can be included in the main context).

Ablation study is missing and thus hard to answer questions such as , is unsupervised learning (i.e., learning in Equation 2) even needed for getting strong results? I would really like to see the performance gains due to unsupervised learning.

Can authors discuss the computation complexity of their method?

Typos:

Variables $T$, $b$ in equation (1) are not defined.

---

> ### Author Response · Authors · 2020-11-15
> **We highly appreciate the valuable comments and we have followed the comments in improving our paper.**
>
> We highly appreciate the valuable and constructive comments of the reviewer. In particular, we have greatly benefited from the comment in terms of making our claims more rigorous and justified.
>
>
> $Q$: Authors argue that the classic graph neural networks which employ graph pooling operations are the bottleneck in identifying necessary substructures (or their interactions) for yielding high discriminative performance. However, such statements are quite loose and need further theoretical justification
>
> $A$: We totally agree with the reviewer that traditional graph pooling (deepsets, sum-pooling or GIN) produces injective mapping, making it a powerful tool for graph classification. It can be risky to claim that there is bottleneck with it in yielding discriminative representations (mixing parts together does not necessarily lose information of the parts). Inspired by the reviewer, we have adjusted our claims throughout the paper to make them more rigorous: our work is just an attempt to introduce well-studied science of complex systems, in particular the idea of explicitly modelling interactions between component parts of a system, in graph classification ("Our Contribution", section 1). To achieve this, we provide a new, identity-preserving graph pooling strategy under the framework of spatial/structural resolutions, and observe encouraging performance gain. **We will not be claiming that traditional graph pooling leads to loss of information**.  In future research, we will pursue a learning theoretic justification of the advantage of explicitly modelling parts and their interactions, besides being potentially easier to interpret.
>
>
> $Q$: It is not clear why right hand side spectrum in Figure 2 will lead to lower generalization performance or over-fitting. It would be great if authors can expand on their explanation.
>
> $A$: The right end of the spectrum in Fig.2 corresponds to learning a dictionary with too many landmarks/code-vectors. As previously discussed in Section 4.1, too many basis may result in overfitting (Marsousi et al., 2014), and a large redundancy of the basis (“mutual coherence”) significantly lowers the chance of a faithful signal recovery or sparse coding/clustering (Donoho & Huo., 2001; Mehta & Gray, 2013). Our empirical study also validated this (Fig.5).  Finally, we speculate computing the landmarks is subtly different than maintaining exact subgraph instances in graphlet/motif. We will study this in more depth in our future research.
>
>
> $Q$: Figure 1 lacks detail description and it is not clear what "all the nodes are mixed into one" means … a general suggestion would be to add more descriptive caption for each Figure
>
> $A$: We have added detailed captions for all the figures as reviewer suggested. “all the nodes mixed into one” means that graph pooling turns a graph of n nodes (or n vectors) into a single vector (by graph pooling).
>
>
> $Q$: I would suggest to provide compelling real-world examples (or do more qualitative analysis) besides the strong empirical performance in the main paper (there is some discussion in appendix but highlights can be included in the main context)."
>
> $A$: We thank the reviewer for this suggestion. As suggested, we have added more discussions and results in Fig.7, Section 5. These new results shed light on how prediction performance is affected by hyper-parameters.  Currently, we are also digging deep into the learned landmarks in molecule/protein classification, to extract concrete examples to demonstrate the usefulness of the learned sub-graph pattern to domain experts in medical research (as inspired by the comment of the reviewer). This requires more chemistry knowledge and we are working on it. Social networks (Collab and IMDB) do not have node attributes so appear less fruitful for such case study.
>
>
> $Q$: Ablation study is missing and thus hard to answer questions such as, is unsupervised learning (i.e., learning in Equation 2) even needed for getting strong results? I would really like to see the performance gains due to unsupervised learning.
>
> $A$: This is an interesting question. As reviewer suggested, we have added ablation studies on the impact of the node2vec loss (2) on the generalization performance (Fig.7(b)). As can be seen, the node2vec does benefit generalization when enforced properly, since it promotes smoothness of embedding w.r.t. sub-structure interactions (as discussed in Section 3.1). We also have added the impact of the clustering loss (4) in Fig.7(c), with similar observation. We speculate that unsupervised losses can serve as regularization to benefit the learning task.
>
>
> $Q$: Can authors discuss the computation complexity of their method?
>
> $A$: Our time complexity is linear with the number of graphs, the average graph size, the average k-hop neighborhood size for each node, and the number of landmarks K. More detailed analysis is in Appendix G.2. We have also added the time consumption in Fig.10 in Appendix.

---

### Decision · Program_Chairs · 2021-01-07
**Final Decision**

**Decision:**

Reject

**Comment:**

The proposed approach seems to have elements of novelty, it is well presented  and  reasonably motivated by the authors. In addition, empirical results seem to be promising. However, although rebuttal helped to clarify some of the pending issues, there are concerns on the fact that the raised issue about "resolution dilemmas" does not find in the paper a quantitative treatment. Without that, it is difficult to fully understand how to drive the learning of useful structural landmarks. Thus, notwithstanding the paper seems to contribute in a significant way to the advancement of the GNN field, it still needs additional work to better develop  the proposed concepts in a quantitative theory.